# Effectiveness and Selectivity of Pre- and Post-Emergence Herbicides for Weed Control in Grain Legumes

**DOI:** 10.3390/plants13020211

**Published:** 2024-01-11

**Authors:** Angeliki Kousta, Christos Katsis, Anastasia Tsekoura, Dimosthenis Chachalis

**Affiliations:** 1Laboratory of Weed Science, Department of Pesticides’ Control and Phytopharmacy, Benaki Phytopathological Institute, 145 61 Kifisia, Greece; a.tsekoura@bpi.gr (A.T.); d.chachalis@bpi.gr (D.C.); 2Laboratory of Plant Breeding and Biometry, Department of Crop Science, Agricultural University of Athens, 118 55 Athens, Greece; xristoskatsis3@gmail.com

**Keywords:** grain legumes, herbicides, herbicide effectiveness, herbicide selectivity, weed management, grain yield, phytotoxicity

## Abstract

Grain legumes represent important crops for livestock feed and contribute to novel uses in the food industry; therefore, the best cultivation practices need to be assessed. This study aimed to identify herbicides to meet the current need for controlling broadleaf weeds without phytotoxicity in the grain legume crop per se. Field experiments were undertaken during the 2019 and 2020 growing seasons and laid out in a randomized complete block design with three replicates as follows: four grain legume crops (vetch, pea, faba bean, and white lupine) and nine pre-emergence (PRE) or post-emergence selective (POST) herbicide treatments (PRE: aclonifen, pendimethalin plus clomazone, metribuzin plus clomazone, benfluralin, terbuthylazine plus pendimethalin, S-metolachlor plus pendimethalin, flumioxazin; POST: pyridate, imazamox) alongside weedy check plots. Plant phytotoxicity, crop dry matter, yield features, weed presence, and weed dry matter were assessed during the experiments. There was differential efficacy among the nine herbicide treatments; the weed control was more effective in the case of *Veronica arvensis* L. and *Sonchus* spp. L. compared with *Chenopodium album* L., *Sinapis arvensis* L., and *Silibum marianum* L. regardless of the herbicide treatment. The most effective PRE herbicide was flumioxazin, which had the greatest control over the majority of weeds (>70%) resulting in the lowest total weed biomass. The second-best treatment was benfluralin and the mixture of terbuthylazine plus pendimethalin (both had only limited control in *S. arvensis*). The best POST herbicide was imazamox, with only limited control in *S. arvensis*. The tested herbicides caused low to medium and transient levels of phytotoxicity mainly in vetch and secondly in peas but not in faba beans and lupines. Concerning all weed management treatments, benfluralin resulted in the highest grain yields for all four grain legume crops during both growing seasons. Among grain legumes, vetch had the highest competitive ability against weeds, whereas peas were the least tolerant against weed competition.

## 1. Introduction

Legumes are the second most economically important crop in worldwide agriculture after cereals and are grown for both forage and grain [1]. In the past, they were understated in European cropping systems because of cereals and non-legume oilseed dominance. However, the great call for high-protein materials for livestock feed, the need for a reduction in European dependence on imported protein, and the opportunities to use legumes in new foods brought legumes back to the forefront of public debate in Europe [2].

It is currently acknowledged that grain legumes can be a popular choice in farming systems since they can contribute to the nutritional security and resilience of agricultural ecosystems [3]. As a valuable source of vitamins, minerals, protein, and dietary fibers, they play a vital role in human and animal diets [4,5,6]. Moreover, grain legumes provide ecosystem services through their biological nitrogen fixation capacity, improving physical soil properties and enriching soil with N [7,8]. Thus, they contribute to the reduction in overdependence on inorganic nitrogen fertilizers. Due to their cultivation mostly in marginal environments, they can also survive in problematic soils and resist abiotic stresses [9].

The underlined significance of grain legumes led the European Union (EU) to undertake various support measures [2]. Nowadays, grain legumes are cultivated in an area of about 81 million ha and produce more than 92 million tons universally [10]. Among grain legumes, peas (*Pisum sativum* L.), faba beans (*Vicia faba* L.), vetch (*Vicia sativa* L.), and white lupine (*Lupinus albus* L.) are considered important crops for several reasons. Peas are the most widely cultivated grain legume in Europe [11]. They are mainly grown for their green pods and consumed as fresh vegetables and cooked green seeds. Faba beans are the fourth most significant cool-season grain legume and are broadly preferred as a green vegetable, while in several countries they are used as feed. Moreover, the production of faba beans has increased yearly by 2% during the past three decades. Concerning the vetch crop, another cool-season member of the Leguminosae family, it is mainly used as animal feed and is also studied for its seeds as a promising new source of starch and a sustainable source of food for humans [12]. White lupine, a multifunctional legume crop, is cultivated for a broad range of uses, from forage for livestock and food for humans to medicinal and pharmaceutical uses [13].

Even though legume production has been rising globally [14], grain legumes are hardly ever chosen by farmers for cultivation as compared to other crops [2]. The main reason for this fact is that grain legume yields are more variable against biotic and abiotic stresses than those of cereals, presenting yield fluctuations due to their indeterminate growth habit and the relative absence of investment in breeding for stress-resistant cultivars [2]. These yield losses would be partially attributed to the lack of farming practices capable of withstanding biotic stresses such as weed infestation.

Grain legumes present a poor competitive ability against grasses and broadleaf weeds [15]. Due to the relatively slow establishment after sowing and the insufficient soil surface coverage shaped by small tendrils and leaflets, grain legumes are unable to effectively suppress weed growth [15]. Thus, weeds pose a serious threat to the sustainability of grain legumes since they strongly compete with the crop for water, sunlight, and nutrients [16].

Given that weed control management is a major challenge for grain legume production, several cultural, mechanical, and chemical practices have long been studied. Increased crop density improved weed control in vining peas [17] and faba beans [18], which significantly raised the seeding cost of the crop. For faba beans, the combination of row spacing and mechanical control led to a similar level of weed control in comparison to herbicide application, while for field peas, some herbicides could be used alongside these other methods [16]. Another notable strategy for optimizing weeding in grain legumes is the use of competitive cultivars against weeds, especially in farming systems where herbicides are avoided [2].

In most grain legume cropping systems, chemical control is mostly preferred by farmers since herbicides are a more reliable, effective, and profitable method of controlling different weed species [19]. In grain legumes, it is well documented that the most important type of herbicide is the pre-emergence (PRE) one [20]. However, PRE herbicides are significantly dependent on rainfall soon after application, and therefore, in semi-arid conditions (i.e., Mediterranean conditions), frequently inconsistent or partial weed control is documented [20]. The post-emergence selective (POST) herbicides are mostly used to control broadleaf weeds; only a few are registered in Europe for a small range of weed species, officially registered for a limited number of crops; they are the following: aclonifen, bentazone, and imazamox [21].

While the benefits of grain legumes for the nutritional security and resilience of agricultural ecosystems are underlined, the best cultivar practices, including weed management, are urgently needed to support high yields and enhanced profitability of the crops. To the best of our knowledge, the design of a weed management program for these crops is still in its infancy, as little information exists, and there is also a lack of registered herbicides for these crops. In this context, the need for research aiming to identify potential herbicides becomes even more imperative in environments affected by large fluctuations in temperature and precipitation. Climatic factors can affect herbicide efficacy and crop safety [20]. Therefore, environments such as the Mediterranean are of high scientific interest for the study of herbicide efficacy and selectivity in grain legume crops.

In this framework, there is a great need to identify herbicides (single or ready-mixed ones) that would provide broadleaf weed control with adequate crop safety. Therefore, this study aimed to (a) measure the efficacy of seven PRE and two POST herbicides against broadleaf weeds and (b) assess the selectivity of these herbicides on the four grain legumes (i.e., peas, faba bean, vetch, and white lupine). In this study, the effects of herbicide treatments on weeds and legumes were studied under semi-arid Mediterranean conditions during two growing seasons.

## 2. Results

### 2.1. Herbicide Effectiveness

Weed species associated with vetch, pea, faba bean, and lupine in the experimental plots were identified. A total of 10 annual and perennial weed species were recorded, and weed biodiversity remained the same between the two growing seasons. The majority were annual broadleaved weed species. Among them, the most dominant species were *Chenopodium album* L., *Sinapis arvensis* L., *Silibum marianum* L., *Sonchus* spp. L., and *Veronica arvensis* L. (Table 1). Weed species identification also revealed the presence of *Fumaria officinalis* L., *Anthemis chia* L., *Falaris minor* Retz., *Stelaria media* L., and *Capsella bursa-pastoris* L. Medik in lower densities.

Aggregated data from all four legume species on the control of the major weed species showed a differential control by individual herbicide treatment (Table 2). In general, it was easier to control *V. arvenis* and *Sonchus* spp. as compared to *C. album*, *S. arvenis*, and *S. marianum* regardless of herbicide treatment in both growing seasons (Table 1). With regard to the control of individual weed species and specifically the control of *C. album*, a high efficacy (>70%) was achieved by flumioxazin, benfluralin terbuthylazine plus pendimethalin, and imazamox, while moderate efficacy (55–69%) was documented for the application of aclonifen. In contrast, pendimethalin plus clomazone, metribuzin plus clomazone, s-metolachlor plus pendimethalin, and pyridate showed low efficacy (<54%). None of the herbicides gave excellent control of *S. marianum*. More specifically, the use of aclonifen, benfluralin, terbuthylazine plus pendimethalin, and flumioxazin resulted in a moderate control of *S. marianum*, while pendimethalin plus clomazone, metribuzin plus clomazone, s-metolachlor plus pendimethalin, pyridate, and imazamox showed poor efficacy. Among the dominant weeds, the most susceptible to the herbicides applied were *V. arvensis* and *Sonchus* spp. All herbicide treatments had a high efficacy (>70%) for both weed species, except for the post-emergence imazamox, which had a medium efficacy. Regarding the weed control of *S. arvensis*, a high efficacy (>70%) was recorded for the use of aclonifen and flumioxazin, while a moderate efficacy (55–69%) was recorded for the application of all the remaining herbicides except for benfluralin and s-metolachlor plus pendimethalin, which had a low efficacy (<54%) (Table 1).

Figure 1 shows the overall weed control based on the visual assessment of each herbicide treatment at 62 days after sowing (DAS). The data showed that all the herbicide treatments resulted in a superior control compared to the weedy check plots. Moreover, the efficacy of each herbicide treatment was not statistically different between the two growing seasons. In particular, pre-emergence applications of either flumioxazin or benfluralin resulted in highly effective (>70%) weed control. In addition, three herbicide treatments (i.e., aclonifen, terbuthylazine plus pendimethalin, imazamox) showed moderate overall weed control, whereas all the other herbicide treatments showed low-efficacy weed control (Figure 1).

All the herbicide treatments resulted in a reduced weed abundance of the major weed species compared to the control plot where no weed control was applied (Table 2). Total weed biomass (TWB) in the weedy check plots was higher in GS1 (257.7 g m^−2^) than in GS2 (192 g m^−2^), although the differences were not statistically significant. All weed control treatments caused a significant reduction in TWB compared to untreated plots in both growing seasons (*p* < 0.05). In this context, the analysis of the TWB data revealed two groups of herbicide treatments, as follows: (a) the one with the highest TWB reduction (i.e., for GS1, values from 24 to 46 g m^−2^ vs. 192 g m^−2^ in the control) related to aclonifen, benfluralin, terbuthylazine plus pendimethalin, flumioxazin, and imazamox, and (b) the one with lower TWB reduction (i.e., for GS1, values from 98 to 109 g m^−2^ vs. 192 g m^−2^ in the control) related to pendimethalin plus clomazone, metribuzin plus clomazone, s-metolachlor plus pendimethalin, and pyridate (Table 2).

### 2.2. Herbicide Selectivity, Crop Biomass, and Grain Yield

#### 2.2.1. Vetch (*Vicia sativa*)

In general, phytotoxicity symptoms were observed in both growing seasons with more pronounced effects in GS2 compared to GS1 (Table 3). In GS1 (November–June 2019–2020), the herbicide mixtures of terbuthylazine plus pendimethalin and s-metolachlor plus pendimethalin caused severe crop injury (<20%) of 20% and 22%, respectively. In addition, the herbicide treatments benfluralin, flumioxazin, and imazamox showed phytotoxicity at the rate of 10–15% (average damage and consistent on vetch), whereas pendimethalin plus clomazone, metribuzin plus clomazone, and pyridate caused inconsistent lower damage of 3–7%.

In GS2 (November–June 2020–2021), more herbicide treatments (i.e. terbuthylazine plus pendimethalin, s-metolachlor plus pendimethalin, pendimethalin plus clomazone, benfluralin, and imazamox) caused 18–25% severe damage on vetch. Furthermore, pyridate and flumioxazine herbicide treatments caused an average of 12% injury, while aclonifen and metribuzin plus clomazone affected vetch with 5–7% inconsistent phytotoxicity. Phytotoxicity symptoms were mainly documented as leaf chlorosis, leaf deformation, reduced growth, and slower vegetative growth, the intensity of which varied between the herbicide treatments. In most cases, there was a gradual overcoming of the phytotoxicity symptoms by the vetch plants.

In both growing seasons, vetch dry biomass was not significantly decreased by herbicide application compared to the weedy check plots in most herbicide treatments (*p* < 0.05), with some exceptions (Table 3). In GS1, only the three herbicide treatments of aclonifen, flumioxazine, and imazamox showed significantly higher plant dry biomass than the weedy check plots (*p* < 0.05). In the following growing season (GS2), only the herbicide treatments of aclonifen, benfluralin, flumioxazin, and imazamox enhanced the vetch growth, resulting in greater plant dry biomass compared to the untreated plots.

At harvest, total vetch grain yield (t ha^−1^) did not differ between the growing seasons regardless of the weed management treatments. In GS1, only three herbicide treatments showed similar values to the weedy check plots (i.e., pendimethalin plus clomazone, terbuthylazine plus pendimethalin, and pyridate), whereas in GS2, six herbicide treatments showed similar values to the weedy check plots (i.e., aclonifen, pendimethalin plus clomazone, s-metalachlor plus pendimethalin, pyridate, flumioxazine, and imazamox).

#### 2.2.2. Pea (*Pisum sativum*)

In general, pea plants showed low phytotoxicity symptoms in both growing seasons, irrespective of the herbicide treatment (Table 4). The highest values were 12% for terbuthylazine plus pendimethalin in GS1 and 18% for s-metolachlor plus pendimethalin in GS2. Also, the treatment of pea plants with terbuthylazine plus pendimethalin caused an average and consistent damage of 12–13% in both seasons, whereas the treatment of pendimethalin plus clomazone caused an average and more consistent damage of 7–8%. All other treatments caused negligible phytotoxicity.

In both growing seasons, in most herbicide treatments, pea dry biomass was not significantly decreased by herbicide application compared to the weedy check plots at 62 DAS (*p* < 0.05) (Table 4). At GS1, only three herbicide treatments (i.e., aclonifen, flumioxazine, and imazamox) showed higher plant dry biomass than the untreated plots, while four herbicide treatments (i.e., aclonifen, benfluralin, flumioxazin, and imazamox) promotively impacted the pea dry biomass in GS2.

At harvest, total pea grain yield (t ha^−1^) did not differ between the growing seasons regardless of weed management treatments. In GS1, only three herbicide treatments showed similar yield values to the weedy check plots (i.e., pendimethalin plus clomazone, terbuthylazine plus pendimethalin, and pyridate), whereas in GS2, six herbicide treatments were measured showed similar yield values to the weedy check plots (i.e., aclonifen, pendimethalin plus clomazone, terbuthylazine plus pendimethalin, s-metalachlor plus pendimethalin, pyridate, flumioxazine, and imazamox).

#### 2.2.3. Faba Bean (*Vicia faba*)

In both growing seasons, faba bean plants showed negligible phytotoxicity in response to all herbicide treatments applied (Table 5). In terms of faba bean growth, the use of most herbicides did not significantly decrease the faba bean dry biomass compared to the weedy check plots (*p* < 0.05) (Table 5). Among the growing seasons, the crop was significantly affected by aclonifen, flumioxazine, and imazamox in GS1 with greater plant dry biomass than in the weedy check plots. Similar effects were observed after the application of aclonifen and benfluralin in GS2.

At harvest, total faba bean grain yield (t ha^−1^) did not differ between the growing seasons regardless of the weed management treatments. In GS1, only three herbicide treatments showed similar yield to the weedy check plots (i.e., pendimethalin plus clomazone, terbuthylazine plus pendimethalin, and pyridate), whereas in GS2, six herbicide treatments (i.e., aclonifen, pendimethalin plus clomazone, s-metalachlor plus pendimethalin, pyridate, flumioxazine, and imazamox) affected grain yield in this manner.

#### 2.2.4. Lupine (*Lupinus albus*)

In both growing seasons, lupine plants showed negligible phytotoxicity in response to all herbicide treatments applied (Table 6). In addition, in most herbicide treatments, plant dry biomass was not significantly decreased by herbicide treatment compared to the weedy check plots (*p* < 0.05) (Table 6). In GS1, only three herbicide treatments (i.e., aclonifen, flumioxazine, and imazamox) resulted in greater plant dry biomass than the untreated plots. For GS2, only two herbicide treatments (i.e., aclonifen, benfluralin) led to increased lupine dry biomass.

At harvest, total lupine grain yield (t ha^−1^) did not differ between the growing seasons regardless of the weed management treatments. In GS1, only four herbicide treatments showed analogous values to the weedy check plots (i.e., pendimethalin plus clomazone, metribuzin plus clomazone, and pyridate), whereas in GS2, six herbicide treatments were recorded as having an effect on lupine grain yield (i.e., aclonifen, pendimethalin plus clomazone, s-metalachlor plus pendimethalin, pyridate, flumioxazine, and imazamox). Lastly, the significantly highest grain yield was observed in plants treated with pre-emergence benfluralin at 4.1 t ha^−1^ and 3.9 t ha^−1^ in the first and second growing seasons (Table 6).

## 3. Discussion

Although grain legumes are important crops for livestock feed and have novel uses in the food industry, best cultivation practices to increase their productivity are still being researched. A literature review demonstrated that weed control in these crops has a serious impact on agronomic and yield features [19,22,23]. Peas weakly suppress weeds, especially during the critical 40–60 days after sowing [24]. Weed infestation can reduce pea yield by up to 64% if no weed control practice is applied [25]. Weed infestation is also a major constraint in faba bean production. This is especially problematic between 25 and 75 days after sowing since weeds can decrease yields by up to 50% [22,26]. In addition, high susceptibility to weeds is also measured in vetch and white lupine [23]. It is necessary to develop integrated weed management (IWM) systems utilizing herbicides that focus on minimizing the yield losses caused by weeds and supporting the profitability of the legume crops.

### 3.1. Herbicide Effectiveness

In the present study, the experimental site was selected due to the high number of weed species (10 species in total). Among these species, five broadleaf weeds were the dominant ones allowing us to test the efficacy of herbicides on the major weed species in the Mediterranean conditions (Table 1). The weed flora of the studied grain legumes remained the same, and similar levels of weed biomass were recorded during both growing seasons.

All herbicide treatments in this study resulted in a reduced presence of the major weed species compared to the control plot where no weed management practice was applied (Table 2). Aggregated data from all four legume species showed a differential control of the dominant weed species per individual herbicide treatment (Table 1). Regarding pendimethalin, an important PRE herbicide in grain legumes, there was a differential efficacy of mixtures containing pendimethalin depending on the type of mixture per weed species. More specifically, the mixture of pendimethalin plus s-metolachlor had the poorest weed control of the three most difficult-to-control weeds (i.e., *C. album*, *S. arvensis*, and *S. marianum*). On the other hand, the mixture of pendimethalin with clomazone improved weed control, whereas the mixture with terbuthylazine significantly increased the control (Table 1). Similar results on the efficacy of pendimethalin applied either alone or in mixtures were also depicted in a previous study by Vasilakoglou et al. (2013) [21]. In particular, pendimethalin was shown to provide the greatest control of *C. album* compared to other herbicide treatments. In addition, Juhasz et al. (2023) reported high control (>70%) of common *C. album* with pendimethalin [23]. Similarly, Chomas and Kells (2004) found that pendimethalin provided high (98%) and more consistent control of *C. album* in maize (*Zea mays* L.) [27]. On the contrary, Karimmojeni et al. (2015) reported that there were weed escapes after pre-emergence pendimethalin applications, requiring one-handed weeding to achieve adequate control of broadleaf weeds [28]. In this context, the control of Brassicaceae weed species (i.e., *S. arvenis*) with herbicides other than pendimethalin tends to be moderate to poor in most Greek growing regions. Other herbicides or mixtures are therefore required [21], a statement that is also consistent with the Australian guidelines for weed control in grain legumes [20].

Concerning other pendimethalin mixtures, Barua et al. (1990) reported that the pre-plant incorporated (PPI) application of pendimethalin plus imazethapyr effectively controlled broadleaf weeds in lentil [29]. Similar results were reported by Ahmadi et al. (2016), who found that tank mixtures of pendimethalin plus imazethapyr suppressed most broadleaf weeds reasonably well [30]. For other mixtures of pendimethalin, mixtures with s-metolachlor did not increase the efficacy of *C. album* control (Table 1). Τhis result is in line with previous reports where similar control (66%) of broadleaf weeds was recorded after the application of s-metolachlor [27].

In addition, aclonifen was shown to be highly effective in controlling all dominant broadleaf weeds in grain legume crops (Table 1). Similar results of high efficacy were reported by Americanos and Droushiotis (1999) in pea and vetch crops in plots treated with aclonifen [31]. Regarding the mixture of metribuzin plus clomazone, it resulted in poor control of the three difficult-to-control weeds (*C. album, S. arvenis,* and *S. marianum*; Table 1) and provided one of the lowest overall weed controls (Figure 1) and one of the highest total weed biomass values (Table 2). Our finding also revealed that compared to the other herbicide treatments, flumioxazin had the greatest control of major weeds (Table 2), the highest overall weed control (Figure 1), and the lowest total weed biomass (Table 3). Most of the above results agreed with the report of Vasilakoglou et al. (2013) [21].

Among the post-emergence herbicides, pyridate caused a moderate control of *C. album* (Table 1). This finding of our study is partly in line with previous studies in grain legumes, which reported good control of the broadleaf weed species in lentil [28] and chickpea crops [32]. As for imazamox, even if it is a registered herbicide in Europe for post-emergence use only in alfalfa crops (*Medicago sativa* L.), it could be an effective herbicide for broadleaf weed control in grain legumes as well. In our study, imazamox had a variable efficacy (i.e., low, moderate, high) depending on the weed species; the high control of *C. album* agrees with the previous report by Vasilakoglou et al. (2013) [21]. However, in one of the most difficult broadleaf weeds to control (i.e., *S. arvensis*), imazamox resulted in poor control (Table 1). The main reason for this poor performance may have been that the specific weed species had emerged later than the herbicide application date and therefore practically escaped the treatment.

### 3.2. Herbicide Phytotoxicity, Plant Growth, and Yield

Overall, some of the herbicides tested in these experiments had low to medium levels of phytotoxicity mainly in vetch and secondly in peas; legible phytotoxicity was measured in faba beans and lupines (Table 3, Table 4, Table 5 and Table 6). The specific symptoms of phytotoxicity were observed as follows, depending on the herbicide used: (a) stunted seedlings and thickened and shortened roots due to pendimethalin and (b) foliar chlorosis and necrosis due to pyridate, s-metolachlor, terbuthylazine, and imazamox. In most cases, some growth retardation due to phytotoxicity was observed, but the symptoms were mostly transient, and plants mostly recovered during the crop development. Previous studies have documented that the application of pendimethalin in regions with low rainfall and organic matter suppresses sensitive legume crops such as lentils [33].

In our study, imazamox resulted in low phytotoxicity in vetch (Table 3), and the symptoms decreased with time due to plant regrowth; these results agree with previous reports [21,23]. However, reports in the literature showed that the selectivity of imazamox for vetch tended to be variable; imazamox treatments caused much higher (33–80%) phytotoxicity at similar doses [34]. Regarding the effects on growth and yield, no negative effects were observed in vetch (Table 2). This finding of our study agrees with previous reports, where no growth or yield penalty was observed even at the high phytotoxicity level [34]. Particular attention should be paid to the indirect effects of imazamox in legumes, especially vetch, due to the high herbicide accumulation in plant tissues. A previous study by García-Garijo et al. (2014) reported high imazamox accumulation in vetch, with concentrations more than six times higher than those detected in beans (*Phaseolus vulgaris* L.) [35]. It is known that high herbicide accumulation in leguminous plants can negatively affect symbiosis and biological nitrogen fixation [36,37], hence affecting growth and yield, particularly under abiotic stress conditions.

Concerning the mixtures of s-metolachlor plus pendimethalin, they resulted in high phytotoxicity levels (Table 3 and Table 4) in vetch and peas in the second growing year, and the symptoms were chlorotic leaves. Although this phytotoxicity was transient, a decrease was measured in the plant growth and yields in the above legume species. Our results related to vetch agreed with those reported by Vasilakoglou et al. (2013) [21]. In addition, the results of our study also revealed that among the post-emergence herbicides, flumioxazin was reported to have the minimum phytotoxicity levels and highest legume plant growth/yield in vetch and pea compared to the other herbicide treatments (Table 3 and Table 4). These results agreed with the research by Vasilakoglou et al. (2013) [21]. Among all weed management treatments, benfluralin resulted in the highest grain yields for all the four grain legume crops in both growing seasons.

Noncontrolled weeds reduced yields differently in the four legume species. More specifically, vetch showed the lowest (but also the most variable) yield reduction, ranging from 11% to 33% (Table 3), followed by faba bean (18–27% reduction; Table 4), lupine (22–28% reduction; Table 6), and finally pea (30–39% reduction; Table 4). The highest competitive ability of vetch against weeds, compared to the other three legume species, could be attributed to its vigorous growth rate during the early stages of growth [38]. On the contrary, peas showed the lowest competitive ability; this result agrees with the results of previous studies [16,39,40,41]. In our study, faba beans and lupines were in between the above species in terms of weed tolerance. Apparently, there is a strong effect between genotype and the environment, as shown by Abou-Khater et al. (2022) [42]. In addition, the four legume crops are rain-fed in Greece, and the growing area (Thessaly region) has an average rainfall of 310 mm, which represents a drought environment to a certain extent. Under these growing conditions, increased competition between crops and weeds is expected due to the limited soil moisture conditions.

## 4. Materials and Methods

### 4.1. Experimental Site

Field experiments were conducted in Central Greece (latitude 39°18′25.28″ N, longitude 22°7′30.19″ E, altitude 131 m above sea level) for two consecutive growing seasons: November–June 2019–2020 and November–June 2020–2021. The soil was sandy loam (sand 36%, clay 24%, and loam 40%) with a pH value of 7.7, organic matter 1.26%, and CaCO_3_ 10.9%. Weather data were collected daily from the meteorological station located near the experimental area. Rainfall and average, minimum, and maximum air temperatures were reported as monthly mean data for the growing season (Figure 2). The weather conditions during the growing seasons GS1 and GS2 were quite different. The average air temperatures during the growing periods did not differ greatly. The total rainfall was higher from November 2019 to July 2020 than in the second growing season. However, the total rainfall of both growing seasons remained lower than the historical mean annual rainfall at the site of 310 mm. These climatic conditions are typical of the Mediterranean basin, where winters are cold and rainy.

### 4.2. Plant Materials and Experimental Design

Grain legumes, pea (*Pisum sativum* L.), faba bean (*Vicia faba* L.), vetch (*Vicia sativa* L.), and lupine (*Lupinus albus* L.), which are widely grown in Greece, were selected for the experiments. The specific characteristics of the four commercial cultivars are as follows: cv. Olympos is an early pea cultivar (Institute of Industrial and Fodder Plants, Larissa, Greece), cv. Tanagra is a mid-late faba bean cultivar (Institute of Industrial and Fodder Plants, Larissa, Greece), cv. Evinos is an early vetch cultivar (Agroland SA Company, Karditsa, Greece), and cv. Multitalia is an early lupine cultivar (Agroland SA Company, Karditsa, Greece). A randomized complete block (RCB) design with three replications was arranged for each species. The experiments included ten treatments: untreated weedy check (control), application of seven different pre-emergence (PRE) and two post-emergence (POST) herbicides, and herbicide mixtures. The weedy check (control) plots received no weed management treatment. The description and doses of the herbicides tested are summarized in Table 7.

All herbicides were sprayed once at the recommended doses with a local-made precision sprayer utilizing an air-pressure system, equipped with flat-fan nozzles (XR TeeJet^®^, TeeJet^®^ Technologies, Glendale Heights, IL, USA), calibrated to deliver 350 L ha^−1^ at 300 kPa. The spraying boom was approximately 50 cm above the plant canopy. The spraying environmental conditions were ideal (no wind, no rain after the application) for the experiment. Pre-emergence herbicides were applied in the period after sowing and before the emergence of grain legume crops. Post-emergence herbicides were applied when most of the weeds were at the 3–4-leaf stage. Water was applied to the untreated control in the same manner as in the other treatments. The total size of each plot was 4 m in length and 1 m in width, and each plot consisted of 4 rows spaced 0.25 m apart.

### 4.3. Crop Management

Seedbed preparation was conducted by plowing at a depth of 20 cm and followed by shallow tillage. Seeds of commercial cultivars were sown separately at a depth of 3 cm on 25 November 2019 and 28 November 2020 to achieve the recommended seed rate of 160 kg ha^−1^ for faba bean and lupine and 112 kg ha^−1^ for pea and vetch. No basal fertilization or disease and pest control practices were followed. Plants were rainfed without any supplemental irrigation during the growing periods. The four grain legume crops reached seed maturity after the same period for the two consecutive growing seasons and were harvested on 22 June 2020 and 15 June 2021.

### 4.4. Sampling and Measurements

The experiment aimed to determine the effect of herbicide treatments on overall, specific weed control and legume plant growth and grain yield. The herbicide efficacy and plant phytotoxicity evaluations were performed 4 weeks after each treatment (WAT) and were assessed by visual rating. A scale of 0 to 100 was used for the herbicide efficacy, where 0 depicted no weed control and 100 depicted the total control of weeds. Based on the visual estimations, three broad control scales were developed as follows: high efficacy (>70% control), moderate efficacy (55–69% control), and low efficacy (<54% control). Scaling was based on the overall presence/growth of the weed species in the whole plot. Regarding plant injury level, it was also evaluated visually on a scale of 0 to 100, where 0 represented no injury of grain legumes and 100 represented complete plant necrosis [43]. More specifically, the classification of crop damage was performed according to the European Weed Research Council (EWRC) standard method as follows: damage or less necrosis (1–3.5%), inconsistent less damage (3.5–7%), average and more consistent damage on crop (7–12.5%), average and consistent damage on crop (12.5–20%), heavy damage on crop (20–30) [44].

Data on the weed population and dry matter were recorded 62 days after sowing (DAS), when most of the weeds had already emerged. Weeds were sampled from three 0.5 m^−2^ quadrants placed on the diagonal of each experimental plot. Aboveground weed biomass was collected, identified, and counted, and then it was oven-dried at 70 °C for 48 h (Binder FD023, binder GmbH, Tuttlingen, Germany) to a constant weight to allow the recording of weed dry weight (Kern FCF 30K-3, KERN & Sohn GmbH, Balingen, Germany). The values obtained (means of three quadrants) were converted per unit area for biomass (g m^−2^).

At harvest time, five plants were randomly selected from the middle rows of each plot, hand-harvested, and threshed using a laboratory thresher (LD350, Wintersteiger, Ried im Innkreis, Austria) in order to assess yield. Seed yield was measured at 13% seed moisture content. After the pods were removed, the aboveground stems and leaves of each harvested plant were cut above the soil, dried at 70 °C for 48 h (Binder FD023, Binder GmbH, Tuttlingen, Germany), and then weighed (Kern FCF 30K-3, KERN & Sohn GmbH, Balingen, Germany) to determine the total dry weight of each plant.

### 4.5. Statistical Analysis

All measured and derived data were subjected to analysis of variance (ANOVA), using the statistical software package Statgraphics Centurion XVI Version (Statgraphics Technologies, Inc., The Plains, VA, USA). Before analysis, the aboveground biomass weight was transformed as a square root to homogenize the variance. The significance of differences among treatments was estimated using Fisher’s least significant difference (LSD) test where probabilities are equal to or less than 0.05 (a = 5%).

## 5. Conclusions

Based on the results of the present work, we observed that specific weeds presented a negative effect due to weed–crop competition on the growth and yields of four grain legumes (i.e., vetch, pea, faba bean, and lupine) since plant dry biomass and yields were reduced in the weedy check plots. All the herbicide treatments (i.e., seven PRE and two POST herbicide treatments) decreased the weed presence of the major weed species compared with the untreated plots. Among the dominant weeds, the most easy-to-control weed species were *V. arvensis* and *Sonchus* spp., whereas the most difficult-to-control weeds were *C. album*, *S. arvensis*, and *S. marianum*. The best PRE herbicide was flumioxazin, which had the greatest control over major weeds and resulted in the lowest total weed biomass, followed by benfluralin and the mixture of terbuthylazine plus pendimethalin (the latter ones had only limited control of *S. arvensis*). The best POST herbicide was imazamox, with only limited control of *S. arvensis*. The herbicides tested caused low to medium levels of phytotoxicity mainly in vetch and secondarily in peas with negligible phytotoxicity in faba bean and lupine. Among the grain legumes, vetch was the most competitive against weeds, whereas peas were the most susceptible. Concerning all weed management treatments, benfluralin resulted in the highest grain yields for all four grain legume crops for both growing seasons. Further studies are needed to document the sustainable use of herbicides in grain legumes, with an emphasis on integration with non-chemical methods to enhance crop profitability and sustainability.

## Figures and Tables

**Figure 1 plants-13-00211-f001:**
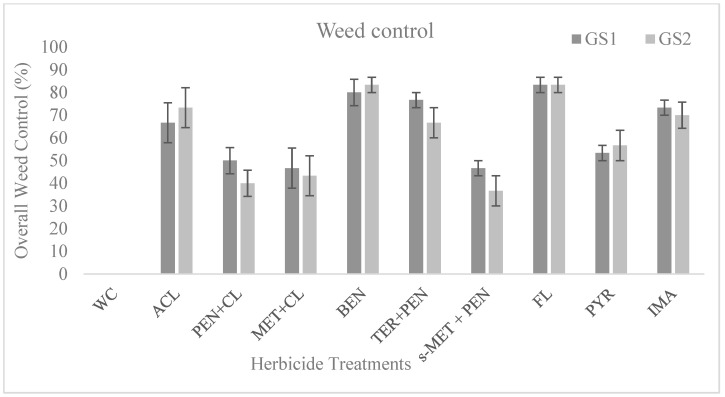
Overall weed control (%) based on the visual scoring of each herbicide treatment at 62 days after sowing (DAS). Visual scoring is as follows: 0% no control, 100% total weed control, during the 2019–2020 (GS1) and 2020–2021 (GS2) growing seasons. Coding of the herbicide treatment is as elsewhere. ACL: aclonifen, PEN + CL: pendimethalin plus clomazone, MET + CL: metribuzin plus clomazone, BEN: benfluralin, TER + PEN: terbuthylazine plus pendimethalin, S-MET + PEN: S-metolachlor plus pendimethalin, PYR: pyridate, FL: flumioxazin, IMA: imazamox plus fatty acid esters plus alkoxylate alcohols–phosphate esters.

**Figure 2 plants-13-00211-f002:**
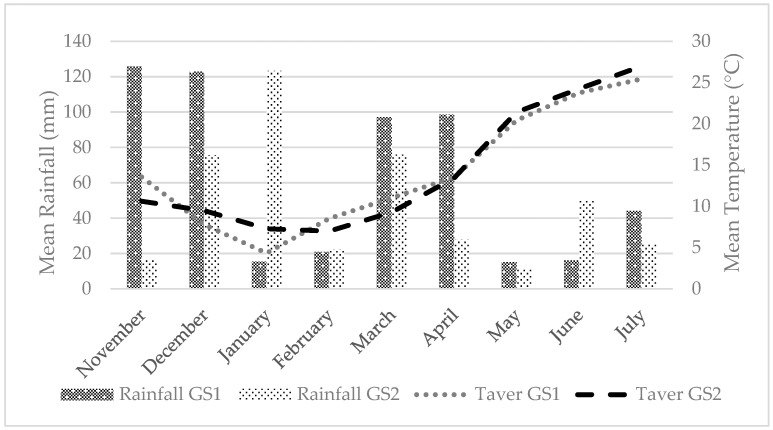
Monthly means of the mean (T_aver_) air temperatures (°C) and total rainfall (mm) for the 2019–2020 (GS1) and 2020–2021 (GS2) growing seasons at the experimental site.

**Table 1 plants-13-00211-t001:** Evaluation of pre-emergence and post-emergence effectiveness (%) on the dominant annual weed species during the 2019–2020 (GS1) and 2020–2021 (GS2) growing seasons; aggregated data from all four legume species (vetch, pea, faba bean, and lupine).

Herbicide Effectiveness on the Dominant Weed Species (%)									
GS1									
Herbicide Treatments									
Weed Species	ACL	PEN + CL	MET + CL	BEN	TER + PEN	s-MET + PEN	PYR	FL	IMA
*Chenopodium album*	66.7	50.0	46.7	80.0	76.7	46.7	53.3	83.3	73.3
*Sinapis arvensis*	70.0	60.0	65.0	51.7	60.0	38.3	15.0	85.0	33.3
*Silibum marianum*	70.0	40.0	45.0	70.0	66.7	28.3	30.0	70.0	51.7
*Veronica arvensis*	85.0	85.0	86.7	88.3	90.0	85.0	81.7	85.0	76.7
*Sonchus* spp.	81.7	83.3	81.7	88.3	85.0	86.7	81.7	83.3	63.3
**GS2**									
*Chenopodium album*	73.3	40.0	43.3	83.3	66.7	36.7	56.7	83.3	70.0
*Sinapis arvensis*	73.3	63.3	63.3	40.0	46.7	55.0	26.7	80.0	30.0
*Silibum marianum*	70.0	45.0	33.3	73.3	70.0	30.0	30.0	76.7	50.0
*Veronica arvensis*	85.0	91.7	86.7	86.7	80.0	86.7	80.0	80.0	75.0
*Sonchus* spp.	80.0	81.7	88.3	86.7	83.3	86.7	81.7	83.3	66.7

ACL: aclonifen, PEN + CL: pendimethalin plus clomazone, MET + CL: metribuzin plus clomazone, BEN: benfluralin, TER + PEN: terbuthylazine plus pendimethalin, S-MET + PEN: S-metolachlor plus pendimethalin, PYR: pyridate, FL: flumioxazin, IMA: imazamox plus fatty acid esters plus alkoxylate alcohols–phosphate esters.

**Table 2 plants-13-00211-t002:** Mean values and analysis of variance for total weed biomass (TWB) of weedy check (WC) plots and herbicide treatment (HT) in 2019–2020 (GS1) and 2020–2021 (GS2) growing seasons. Coding of the herbicide treatment is as elsewhere.

TWB (g m^−2^)		
Herbicide Treatments	GS1	GS2
WC	192 ^a^	257.7 ^a^
ACL	46.3 ^b^	53.7 ^b^
PEN + CL	98.3 ^c^	104 ^c^
MET + CL	105.6 ^c^	112.3 ^c^
BEN	25 ^b^	28.3 ^b^
TER + PEN	40 ^b^	45.3 ^b^
s-MET + PEN	108.7 ^c^	102.7 ^c^
PYR	108 ^c^	110.3 ^c^
FL	23.7 ^b^	48 ^b^
IMA	34.7 ^b^	66.7 ^bc^
Mean	78.2	92.9
LSD_HT (0.05)_	11.3	46.7

ACL: aclonifen, PEN + CL: pendimethalin plus clomazone, MET + CL: metribuzin plus clomazone, BEN: benfluralin, TER + PEN: terbuthylazine plus pendimethalin, S-MET + PEN: S-metolachlor plus pendimethalin, PYR: pyridate, FL: flumioxazin, IMA: imazamox plus fatty acid esters plus alkoxylate alcohols–phosphate esters. Values without a common letter are statistically significant according to LSD (0.05).

**Table 3 plants-13-00211-t003:** Evaluation of pre-emergence and post-emergence selectivity applied to vetch (*Vicia sativa*) plants during the 2019–2020 (GS1) and 2020–2021 (GS2) growing seasons. Phytotoxicity (%): plant susceptibility to herbicide on a scale of 1:100, where 1 depicts no injury on the crop and 100 depicts complete plant necrosis. Plant dry biomass (t ha^−1^) and grain yield (t ha^−1^). Coding of the herbicide treatment is as elsewhere.

GS1			
Herbicide Treatment	Phytotoxicity (%)	Plant Dry Biomass (t ha^−1^)	Grain Yield (t ha^−1^)
WC	0.0	5.4 ^a^	0.9 ^a^
ACL	0.0	6.4 ^c^	1.2 ^b^
PEN + CL	8.3	5.4 ^a^	0.9 ^a^
MET + CL	3.3	5.3 ^a^	1.1 ^b^
BEN	10.0	5.7 ^ac^	1.2 ^b^
TER + PEN	21.7	4.5 ^a^	1.0 ^ab^
s-MET + PEN	20.0	4.4 ^a^	1.2 ^b^
PYR	6.7	5.2 ^ab^	1.0 ^ab^
FL	11.7	6.5 ^c^	1.2 ^b^
IMA	15.0	6.3 ^c^	1.1 ^b^
Mean	-	5.5	1.07
LSD_HT (0.05)_	-	0.27	0.086
**GS2**			
WC	0.0	4.5 ^a^	0.8 ^a^
ACL	6.7	6.3 ^b^	1.1 ^ab^
PEN + CL	18.3	5.2 ^a^	0.9 ^ab^
MET + CL	5.0	5.5 ^ab^	1.1 ^b^
BEN	25.0	6.4 ^b^	1.2 ^b^
TER + PEN	21.7	4.7 ^a^	1.1 ^b^
s-MET + PEN	23.3	4.5 ^a^	1.0 ^ab^
PYR	11.7	4.6 ^a^	0.9 ^ab^
FL	11.7	6.3 ^b^	1.1 ^ab^
IMA	23.3	6.3 ^b^	1.1 ^ab^
Mean	-	5.43	1.03
LSD_HT (0.05)_	-	0.36	0.087

ACL: aclonifen, PEN + CL: pendimethalin plus clomazone, MET + CL: metribuzin plus clomazone, BEN: benfluralin, TER + PEN: terbuthylazine plus pendimethalin, S-MET + PEN: S-metolachlor plus pendimethalin, PYR: pyridate, FL: flumioxazin, IMA: imazamox plus fatty acid esters plus alkoxylate alcohols–phosphate esters. Values without a common letter are statistically significant according to LSD (0.05).

**Table 4 plants-13-00211-t004:** Evaluation of pre-emergence and post-emergence selectivity applied to pea (*Pisum sativum*) plants during the 2019–2020 (GS1) and 2020–2021 (GS2) growing seasons. Phytotoxicity (%): plant susceptibility to herbicide on a scale of 1:100, where 1 depicts no injury on the crop and 100 depicts complete plant necrosis. Plant dry biomass (t ha^−1^) and grain yield (t ha^−1^). Coding of the herbicide treatment is as elsewhere.

GS1			
Herbicide Treatment	Phytotoxicity (%)	Plant Dry Biomass (t ha^−1^)	Grain Yield (t ha^−1^)
WC	0.0	6.7 ^a^	2.3 ^a^
ACL	0.0	8.0 ^b^	3.1 ^b^
PEN + CL	6.7	6.7 ^a^	2.3 ^a^
MET + CL	1.7	6.6 ^a^	3.0 ^b^
BEN	0.0	7.1 ^ab^	3.3 ^b^
TER + PEN	11.7	5.6 ^c^	2.8 ^ab^
s-MET + PEN	8.3	5.5 ^c^	3.1 ^b^
PYR	1.7	6.5 ^ac^	2.6 ^ab^
FL	0.0	8.1 ^b^	3.2 ^b^
IMA	0.0	7.9 ^b^	3.1 ^b^
Mean	-	6.9	2.9
LSD_HT (0.05)_	-	0.3	0.2
**GS2**			
WC	0.0	5.6 ^a^	2.2 ^a^
ACL	0.0	7.9 ^b^	2.9 ^ab^
PEN + CL	8.3	6.5 ^a^	2.5 ^ab^
MET + CL	3.3	6.8 ^ab^	3.0 ^b^
BEN	1.7	8.0 ^b^	3.2 ^b^
TER + PEN	13.3	5.9 ^a^	3.1 ^b^
s-MET + PEN	18.3	5.6 ^a^	2.7 ^ab^
PYR	1.7	5.8 ^a^	2.5 ^ab^
FL	1.7	7.8 ^b^	2.8 ^ab^
IMA	1.7	7.8 ^b^	2.9 ^ab^
Mean	-	6.8	2.8
LSD_HT (0.05)_	-	0.4	0.2

ACL: aclonifen, PEN + CL: pendimethalin plus clomazone, MET + CL: metribuzin plus clomazone, BEN: benfluralin, TER + PEN: terbuthylazine plus pendimethalin, S-MET + PEN: S-metolachlor plus pendimethalin, PYR: pyridate, FL: flumioxazin, IMA: imazamox plus fatty acid esters plus alkoxylate alcohols–phosphate esters. Values without a common letter are statistically significant according to LSD (0.05).

**Table 5 plants-13-00211-t005:** Evaluation of pre-emergence and post-emergence selectivity applied to faba bean (*Vicia faba*) plants during the 2019–2020 (GS1) and 2020–2021 (GS2) growing seasons. Phytotoxicity (%): plant susceptibility to herbicide on a scale of 1:100, where 1 depicts no injury on the crop and 100 depicts complete plant necrosis. Plant dry biomass (t ha^−1^) and grain yield (t ha^−1^). Coding of the herbicide treatment is as elsewhere.

GS1			
Herbicide Treatment	Phytotoxicity (%)	Plant Dry Biomass (t ha^−1^)	Grain Yield (t ha^−1^)
WC	0.0	5.4 ^a^	2.9 ^ab^
ACL	0.0	6.4 ^b^	3.5 ^bc^
PEN + CL	1.7	5.4 ^a^	2.6 ^a^
MET + CL	3.3	5.3 ^a^	3.4 ^bc^
BEN	0.0	5.7 ^ab^	3.7 ^c^
TER + PEN	0.0	4.5 ^c^	3.2 ^abc^
s-MET + PEN	0.0	4.4 ^c^	3.5 ^bc^
PYR	3.3	5.2 ^ac^	3 ^abc^
FL	0.0	6.5 ^b^	3.6 ^bc^
IMA	1.7	6.3 ^b^	3.5 ^bc^
Mean	-	5.5	3.3
LSD_HT (0.05)_	-	0.3	0.3
**GS2**			
WC	0.0	4.5 ^a^	2.5 ^a^
ACL	3.3	6.3 ^b^	3.2 ^ab^
PEN + CL	0.0	5.2 ^a^	2.9 ^ab^
MET + CL	5.0	5.5 ^ab^	3.4 ^bc^
BEN	0.0	6.4 ^b^	3.5 ^b^
TER + PEN	0.0	4.7 ^a^	3.4 ^b^
s-MET + PEN	0.0	4.5 ^a^	3.0 ^ab^
PYR	1.7	4.6 ^a^	2.8 ^ab^
FL	0.0	6.3 ^b^	3.2 ^ab^
IMA	1.7	6.3 ^b^	3.2 ^ab^
Mean	-	5.4	3.1
LSD_HT (0.05)_	-	0.4	0.3

ACL: aclonifen, PEN + CL: pendimethalin plus clomazone, MET + CL: metribuzin plus clomazone, BEN: benfluralin, TER + PEN: terbuthylazine plus pendimethalin, S-MET + PEN: S-metolachlor plus pendimethalin, PYR: pyridate, FL: flumioxazin, IMA: imazamox plus fatty acid esters plus alkoxylate alcohols–phosphate esters. Values without a common letter are statistically significant according to LSD (0.05).

**Table 6 plants-13-00211-t006:** Evaluation of pre-emergence and post-emergence selectivity applied to lupine (*Lupinus albus*) plants during the 2019–2020 (GS1) and 2020–2021 (GS2) growing seasons. Phytotoxicity (%): plant susceptibility to herbicide on a scale of 1:100, where 1 depicts no injury on the crop and 100 depicts complete plant necrosis. Plant dry biomass (t ha^−1^) and grain yield (t ha^−1^). Coding of the herbicide treatment is as elsewhere.

GS1			
Herbicide Treatment	Phytotoxicity (%)	Plant Dry Biomass (t ha^−1^)	Grain Yield (t ha^−1^)
WC	0.0	5.4 ^a^	3.2 ^ab^
ACL	0.0	6.4 ^b^	3.9 ^bc^
PEN + CL	1.7	5.4 ^a^	2.8 ^a^
MET + CL	1.7	5.3 ^a^	3.7 ^abc^
BEN	0.0	5.7 ^ab^	4.1 ^c^
TER + PEN	1.7	4.5 ^c^	3.5 ^abc^
s-MET + PEN	3.3	4.4 ^c^	3.9 ^bc^
PYR	5.0	5.2 ^ac^	3.3 ^abc^
FL	0.0	6.5 ^b^	3.9 ^bc^
IMA	0.0	6.3 ^b^	3.8 ^bc^
Mean	-	5.5	3.6
LSD_HT (0.05)_	-	0.3	0.3
**GS2**			
WC	0.0	4.5 ^a^	2.7 ^a^
ACL	3.3	6.3 ^b^	3.5 ^abc^
PEN + CL	1.7	5.2 ^a^	3.2 ^ab^
MET + CL	1.7	5.5 ^ab^	3.7 ^b^
BEN	0.0	6.4 ^b^	3.9 ^b^
TER + PEN	0.0	4.7 ^a^	3.8 ^b^
s-MET + PEN	3.3	4.5 ^a^	3.3 ^ab^
PYR	0.0	4.6 ^a^	3.1 ^ab^
FL	0.0	6.3 ^b^	3.5 ^ab^
IMA	0.0	6.3 ^b^	3.6 ^ab^
Mean	-	5.4	3.4
LSD_HT (0.05)_	-	0.4	0.3

ACL: aclonifen, PEN + CL: pendimethalin plus clomazone, MET + CL: metribuzin plus clomazone, BEN: benfluralin, TER + PEN: terbuthylazine plus pendimethalin, S-MET + PEN: S-metolachlor plus pendimethalin, PYR: pyridate, FL: flumioxazin, IMA: imazamox plus fatty acid esters plus alkoxylate alcohols–phosphate esters. Values without a common letter are statistically significant according to LSD (0.05).

**Table 7 plants-13-00211-t007:** Characteristics of the herbicides used in the experiment.

Common Name	Trade Name	Mechanism of Action	Time of Application	Recommended Rate (g ai ha^−1^)	Manufacturer
Weedy Control (WC)	-		-	-	-
Aclonifen (ACL)	Challenge 600 SC	Carotenoid biosynthesis inhibitor	PRE	1222	Bayer
Pendimethalin + Clomazone (PEN + CL)	Bismark CS	Carotenoid biosynthesis inhibitor + mitosis inhibitor	PRE	412.5 + 82.5	Sipcam Hellas
Metribuzin + Clomazone (MET + CL)	Metric ZC	Photosystem II inhibitor	PRE	349.5 + 90	UPL
Benfluralin (BEN)	Bonalan 180 EC	Mitosis inhibitor	PRE	1080	Elanco Hellas
Terbuthylazine + Pendimethalin (TER + PEN)	Axion Combi ZC	Photosystem II inhibitor + carotenoid biosynthesis inhibitor	PRE	1000 + 500	Sipcam Hellas
S-metolachlor + Pendimethalin (S-MET + PEN)	Dual Gold 96 EC + Stomp 330 EC	Mitosis inhibitor + carotenoid biosynthesis inhibitor	PRE	960 + 1320	Syngenta
Flumioxazin (FL)	Pledge 50 WP	Protoporphyrinogen oxidase inhibitor (PPO)	PRE	150	Hellafarm
Pyridate (PYR)	Lentagran 45 WP	Photosystem II inhibitor	POST	900	UPL
Imazamox (IMA) + (fatty acid esters + alkoxylate alcohols–phosphate esters)	Pulsar 4 SL + Dash	Acetolactate synthase (ALS)	POST	50 + (375 + 225)	BASF

## Data Availability

The data presented in this study are available on request from the corresponding author. The data are not publicly available due to privacy restrictions.

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
