# Peer review of "Effectiveness and Selectivity of Pre- and Post-Emergence Herbicides for Weed Control in Grain Legumes"

_plants, 2024, doi:10.3390/plants13020211_

Round 1
Reviewer 1 Report
Comments and Suggestions for Authors
The article presents a very interesting topic, I highly appreciate the execution of the field experiment focused on the effect of chemical control of selected types of legumes and weed infestation. The limited originality of the topic and the evaluation of phytotoxicity only by a subjective method are weak points of the article. I understand the importance of the topic for agricultural practice, however, it is still necessary to consider what contribution the article brings to science.
Specific comments:
· Line 8 – delete „2“
· The font used in the abstract and in section Conclusions seems to be mixed – unify it to Palatino style required by MDPI
· Llines 20-22 …. What is meant by „was easier“?
· Line 23 – „major“ should be replaced by „majority of“
Introduction
· Formulate a general scientific hypothesis. The phytotoxicity of herbicides occurring in legumes and the relationship of the Mediterranean climate is an interesting combination.
· Line 47 „problem“ should be replaced by „problematic“
Results
· Number of decimal spaces should be consolidated in Table 2, Table 3, Table 4
M+M
· Is it not clear how heterogeneity of weed infestation was factored in the field experiment?
· Can you confirm if only 10 weed taxa were found in the experiment? Can you share the other 5 latin names of the rest of the found weed taxa?
· Please fill in the dates of sowing and harvesting for individual legumes
Conclusions
· What impact had the herbicide phytotoxicity on the yields of grain legumes?
References
· [26, 27, 28, 30, 33] observe the text format of scientific plant names „Vicia Faba L.“ – „Vicia faba L.“; „Chenopodium Album“ – „Chenopodium album“; „Lens Culinaris“ – „Lens culinaris“; „Triticum Aestivum“ – „Triticum aestivum“; „Pisum Sativum“ – „Pisum sativum“
Reviewer 2 Report
Comments and Suggestions for Authors
This is an experiment on herbicide screening. 4 grain legumes were evaluated using 9 PRE/POST herbicides including their control effect and yield effect etc.
Major concern:
LINE 476: Herbicide test experiments usually require that the area of the plot should not be less than 20 square meters, and this experiment only 4 square meters, which is not in line with the experimental protocol.
Round 2
Reviewer 1 Report
Comments and Suggestions for Authors
It's good to see that you appreciate all the comments.
Comments have been carefully addressed.
Specific comments:
Line 383: „and“ – normal font, not italics
Lines 410 – 659: edit line spacing
References:
· [26, 27, 28, 30, 33] for the name of the plant species, the first letter is lowercase „Faba L.“ – „faba L.“; „Album“ – „album“; „Culinaris“ – „culinaris“; „Aestivum“ – „aestivum“; „Sativum“ – „sativum“
Author Response
Thank you very much for taking the time to review this manuscript. All specific comments on text and references were checked and modified according to your suggestions.
Specific comments:
Line 383: „and“ – normal font, not italics
Response: Checked and modified
Lines 410 – 659: edit line spacing
Response: Checked and modified
References:
- [26, 27, 28, 30, 33] for the name of the plant species, the first letter is lowercase „Faba“ – „faba L.“; „Album“ – „album“; „Culinaris“ – „culinaris“; „Aestivum“ – „aestivum“; „Sativum“ – „sativum“
Response: Checked and modified